# Older people's perceptions of the impact of Dance for Health sessions in an acute hospital setting: a qualitative study

Hilary Bungay ![ORCID] , Suzanne Hughes

► Prepublication history and additional materials for this paper is available online. To view these files, please visit the journal online (http://dx.doi.org/10.1136/bmjopen-2020-044027).

Allied and Public Health, Anglia Ruskin University, Cambridge, UK

**Correspondence to**
Dr Hilary Bungay;
hilary.bungay@anglia.ac.uk

## ABSTRACT

**Objectives** To explore the perceptions of hospitalised older people and their relatives of the impact of taking part in group dance sessions in an acute hospital setting.
**Design** A qualitative descriptive approach was used.
**Setting** An acute hospital trust in the East of England.
**Participants** Purposive sample of dance group participants.
**Intervention** Weekly music and movement sessions for frail older people on Department of Elderly Medicine Wards and the Stroke Rehabilitation Unit. Sessions take place in the ward and are facilitated by a dance artist supported by ward staff.
**Results** Twenty-one semistructured interviews were conducted with older people and/or their relatives. Thematic analysis of the data identified three overarching themes: (1) dance as a physical activity, (2) dance as an opportunity for social interaction and (3) the dance group as a source of emotional support. Sessions were valued as an enjoyable way to undertake physical activity and provided an opportunity for social interaction between patients. This is important as loneliness and boredom are a common occurrence during hospitalisation and are detrimental to overall health and well-being. Patients reported an emotional impact though taking part; happiness from engaging with the group and the release of pent up emotions through the triggering of memories by the music and conversations within the group.
**Conclusion** Dance for Health provides a range of physical, social and emotional benefits for hospitalised older people. Further research is required to investigate the effectiveness of group dance sessions in increasing physical activity on an acute ward and the potential psychological benefits for hospitalised older people.

## INTRODUCTION

When older people are hospitalised, they experience a loss of autonomy, their quality of life can be adversely affected, and they may experience functional decline due to their ability to move around being restricted or limited due to ill health. During hospitalisation, older adults may spend 83% of their hospital stay in bed[1] and this lack of mobilisation may result in 'deconditioning

### Strengths and limitations of this study

► The first study reporting older people's perspectives on the impact of attending a dance group in an acute hospital setting.
► Only patients who attended the groups were interviewed, the reasons for patients choosing not to join the groups were not captured.
► The study took place in one large teaching hospital in the East of England, this may limit transferability to other institutions and cultures.

syndrome'.[2] The hospital environment discourages physical activity with much of the care taking place at the hospital bedside.[3] In the UK recognition of this has led to recent initiatives such as 'End PJ Paralysis' (where PJ stands for Pyjamas), with patients being encouraged to get out of bed and into their clothes to reduce excessive bed rest.[4] The restricted opportunity for movement can also mean that there is little opportunity for engagement in social interaction. This can lead to boredom, a common occurrence in hospitalised older adults, resulting in low mood[5] and cognitive decline.[6]

Dance is a physical activity that not only employs motor skills but also engages cognitive functions, including perception, emotions and memory.[7] Research on dance with older people has been conducted in a range of settings and across the globe.[8–14] But there are a limited number of studies that include frail or functionally impaired adults.[15] Furthermore, not all studies have identified positive outcomes. A randomised controlled trial looking at social dancing and the incidence of falls in older people found that dance participants experienced a higher rate of falls than the control group.[14] There are also methodological weaknesses in some of the existing research with small sample sizes and heterogeneity of tools and scales

used in quantitative studies and a lack of transparency in the reporting of qualitative data analysis.[8]

A large hospital trust in the East of England introduced a weekly *Dance for Health* programme. The underpinning philosophy for *Dance for Health* is that it is about dance and enabling people to be creative and expressive rather than being just an exercise programme or a therapeutic intervention. The sessions are facilitated by a dance artist and take place weekly on wards in the Department of Elderly Medicine and the Stroke and Rehabilitation Unit. The sessions are open to all inpatients on the wards where the programme takes place including people with Parkinson's Disease, delirium, unstable diabetes, stroke, falls and dementia. A description of the format of the sessions is presented in the online supplemental file 1.

## METHOD
### Research design
A qualitative exploratory descriptive approach within an interpretive framework using semistructured interviews was used to explore patients' perspectives and experiences of participating in *Dance for Health.* The study data presented here formed part of a larger service evaluation that adopted a mixed methods approach, including structured observation and interviews with healthcare professionals.[16 17]

A pragmatic approach to sample size and interview method was adopted, taking into account the range of data collection methods in the study, the potential diversity of participants, demands on the participants and the local context.[18] Purposive sampling was used to ensure only patients and/or relatives who had attended the dance sessions were included in the study. Hospital staff identified potential participants and introduced them to the evaluation team. Inclusion criteria included one or more attendance at a dance session and the ability to provide consent. Patients or relatives unable to consent to interview or who had not attended a session were excluded from the research. Participants received a participant information sheet and provided written consent to be interviewed. HB or SH observed the dance sessions as part of the wider study and so were recognised by the participants. They had no personal or professional contact with the patients beyond those sessions. The research team was not informed if potential participants had refused to be interviewed.

### Data collection
To answer the objectives of the study funders and dance programme delivery team, participants were asked about their likes/dislikes about the sessions and the effects they felt taking part had on their overall well-being and physical health (see online supplemental file 2 interview guide). The interviews were semistructured and conducted face to face and the length ranged from 5 min to 33 min. Most interviews took place following dance sessions, except four which took place the following day. Interviews were

conducted in the day room where the activity had taken place or in the ward bay. The interviews and data analysis were undertaken by HB and SH. HB (she) is an associate professor in health and well-being and has undertaken qualitative research with patients, service users and staff across a range of health and community settings over the last 20 years. SH (she is an experienced postdoctoral research associate who has undertaken qualitative research with patients, service users and informal carers. The interviews were audio-recorded and transcribed verbatim for analysis. Transcripts were anonymised using pseudonyms and checked for accuracy against the original recording.

### Data analysis
The data analysis process followed an interpretive approach, and the data were explored to ensure the analysis reflected the accounts of the participants and responded to new themes as identified. Thematic analysis followed Clarke and Braun's six stages of analysis and was conducted systematically to ensure rigour and trustworthiness.[19] To increase dependability and credibility, the first three interviews were listened to by both authors for consistency before further interviews were undertaken. To further ensure dependability, transcripts were checked by HB for accuracy with the audio recordings. This also familiarised the authors with the data. To ensure trustworthiness of analysis both authors independently read each interview transcript and made notes of potential codes, and through discussion, a coding framework was created. The framework was applied across the data set with detailed coding by HB and the coded data was then searched for patterns to develop subthemes which were then clustered and organised into the final themes by HB and SH. The table in online supplemental file 2 provides an example of the coding process and the development of a subtheme and theme.

## RESULTS
Twenty-one patients and/or relatives were interviewed by either HB or SH, who had also attended sessions and undertaken structured observations. Two of those interviewed were men, and the age of those interviewed ranged from 60s to 92s. Participants were not asked about the reasons for their admission to hospital or their ages, but some volunteered their age during the interview and the ages of other participants were estimated (by decade) by the team during the interview interaction.

From the analysis, three main themes related to the perceived impact of the sessions were identified: dance as a physical activity, dance as an opportunity for social interaction and the dance group as a source of emotional support (see figure in online supplemental file 2). Where relevant the reported findings are supported with observations from the sessions. The names used throughout are pseudonyms to protect the identity of the participants.

## Dance as a physical activity

Patients participating in the sessions were elderly and mostly frail, with restricted mobility due to their physical health. The majority remained seated throughout the dance sessions, moving their bodies to varying degrees to the music. Nearly two-thirds (13) of those interviewed, when asked to describe what they liked about the sessions, referred to the physical impact and the opportunity for activity the sessions provided. The dance sessions were seen to be beneficial in terms of loosening up stiff muscles and joints, with recognition that being in bed for long periods of time leads to the body becoming stiff and resistant to movement.

> Yes, everything. Yes, because it helps you to move your limbs and get you going. As I said, I've got no use- my hands lock, and you drop things sometimes. The exercise helps your hands, everything…… Yes, get your body going. (Ellen)

> It just helps everything. Helps the muscles, and the mind and everything….it just relaxes you (Angela)

The music that accompanies the sessions encourages movement, and participants would sway in time to the music and mirror the movements of the dance artist and others in the group. Many of the participants would improvise their own moves with others copying these actions. Even patients who were very frail and unable to make extensive movements would adjust their posture and would sit up being noticeably more upright in their chairs. For some, upbeat music encouraged rhythmic movement, while others responded to slower tracks by stretching their limbs.

> You could think with the music and you could develop your own system of stretches, it sort of guided you into the movements you see. Yes, the music guided you into different things to think of the areas that needed stretching, you see, the bits that hurt. Some of the music gave you the impetus to move and stretch……. You are doing like you are in the water, like you are swimming, that you're in freestyle. (Carol).

While some attributed their movements in the sessions to the music that was played, it was also seeing others in the group move, which inspired them to try and move more.

> Yes, power in it, in the music. To be honest with you, it is the music that makes you want to dance, you know? (Maureen)

> It's uplifting, and it's just rewarding, and movement was great because I fell on my right side of my body. So, just moving my arms slowly was fantastic, but I met the other patients there, not all from (ward 1), some from (ward 2). It was lovely, and I thought, you know, I just feel horrible, and I feel I'm the only sad person in here, and you look at other people, and some people are worse. I'm really grateful. Not grateful that I saw them, but these other people are struggling as

well… Looking at other people and thinking, Well, you know what? I can do this and I can do that. (Alex)

> Wanting to do something. Even just standing up in the chair, suddenly, was a massive thing for people, but they wanted to do it, because they felt motivated, inspired, feeling a bit alive, by the whole situation. (Relative 2)

This relative also went on to suggest that the sessions were like a form of physiotherapy without participants realising that they were engaging in physiotherapy.

It was interesting that those interviewed referred to how they felt other patients benefited from taking part in the sessions. This mostly not only related to enjoyment but also the benefits of the physical movement.

> Oh lots of ways, but it helps them to get moving and moving their limbs, and I've seen (name) lifting her legs up and things like that (Ellen)

> Everyone was able to do the movements that you gave, and going from the sublime to a bit more jazzy it was beautiful. (Mary)

Three of those interviewed referred specifically to exercise, including Collette who called the movements' 'exercises'.

> I think, everybody joined in and were very happy, and they did their exercises, which is lovely. It was very, very nice.

## Dance as an opportunity for social interaction

While the beneficial physical impact of the sessions was acknowledged by participants, the sessions were also valued for the opportunity for the social interactions the sessions provided. The dance activity takes place in a circle formation, so people are sitting close to each other and can see others in the group. Often a sense of fun was apparent, as many of the interactions generated laughter within the group. Such group interactions had a positive effect on those interviewed…

> I thought it was very therapeutic…. Something up here made you feel you were with it and you felt involved. That's how I felt. I didn't feel lonely or shut out. I felt it was company (June)

> We were laughing and talking as well, and I think that helped too. It generated a community feeling, do you know what I mean? (Margaret)

Others remarked that it had given them an opportunity to get to know other people on the ward and the two men particularly valued this.

> I found the session today, which I was a bit doubtful about to start with, very interesting. Because to start with, I learnt the names of patients who were in the same ward as me but who you don't see when you're lying in bed all day….I learnt their names and I found

the music had an enlivening effect on everybody….
(David)

Even those who were perhaps more disabled than others. So I thought it was a very good session. I thought we bonded together….() Well, you're lying in bed and when you can't get up and walk around, you can't talk to people. (Lionel)

The impact of the session extended beyond the allotted time for the session. In one ward, three women who were taking part in the session just before lunch struck up a conversation and decided to stay in the day room following the session to have their lunch together. One of the relatives interviewed also described how, following a session, people in the ward bay who had joined the session were talking about it during the rest of the day. She also remarked how they found common ground because of the session and the mood was buoyant because of joining in. Another relative who was interviewed with her mother also commented on how it was good that the patients could get together in a fun activity.

I thought it was brilliant. And I think it's so good to get the people together as well because they interact with each other and they do things together and it's fun. I thought it was brilliant. (Relative 1)

This same relative also commented on how she liked the fact that she was able to do the session with her mother, and how it gave them something to do together, which was increasingly unusual as her mother aged.

The dance artist would ask people if they would like to hear a piece of music and this would often trigger conversations as people requested certain tracks. In addition, when people recognised certain tunes, this could evoke memories that initiate further conversations and laughter in the group. Through these conversations, people got to know each better and, in some cases, they found that they had previously been at the same school or lived in the same road many years ago. As well as providing a sense of community for individuals who experienced loneliness at home, the sessions gave people the opportunity to make new friends

I enjoyed being with the people, different people (Margaret)

It's fantastic being with other people and just chat with other people (Alex)

Because I am registered disabled, so don't go a lot. I used to be sociable, and I have lost all my friends now, so this is really nice (Polly)

### The dance groups as a source of emotional support
The sessions were often happy and relaxed with conversations and different songs generating amusement and laughter. However, for some participants, the music triggered sadness and sometimes tears, although for others, the tears were brought on through happiness. One such case was observed when a patient said at the end of a

song '*it brought tears to my eyes… I love togetherness, sharing happiness'* in another example when dancing to 'Love me Tender' one patient cried and said '*but I enjoyed it … it really touched me'.* One of the men who was interviewed reflected on how the songs reminded him of his youth, making him feel sad.

I felt a bit tearful when we played songs that took me back to my youth, really. (Lionel)

Although some patients did become sad when certain songs were played, this was not necessarily a negative impact. For example, Maud said that since she had been in the hospital, she had felt that she had had a lump in her chest because she was bottling her emotions and concerns about what was wrong with her. She went on to explain that she had cried in the music and dance session, experiencing a sense of release and feeling that it was safe to cry in front of others who were going through the same thing as herself.

It was very uplifting, and you know. Tearful for me… Everything bottles up, I think. Coming in hospital, and it sort of brings it out. And its cleansing. I feel better for it….because I kind of had a lump here, that it was bottling up…I didn't know what the outcome of that was going to be. So, I just found it very, as I said, uplifting and healing. I don't know. I think lots of stuff goes on in your head when you come into hospital. When you don't know really what's wrong…. but it's a worrying time for the family. Yes, bottled up, and you know. But being able to talk about it. And cry, you know, even cry in silence. It's just cleansing. Being here, yes. Among other people who are going through the same, or more or less the same things, you know. (Maud)

If participants were upset the rest of the group were always very supportive and offered words of encouragement, this included touching the person's arm or taking hold of a hand. Such actions are a form of emotional support that is provided by a being part of group and can help people to cope with stressful situations.

Most participants, however, appeared to really enjoy the sessions and spoke about how much they had enjoyed the dance and found it uplifting,

and I really enjoyed that. It really lifted up my spirits. Honestly, I love it. I've been feeling so low, and it just helped me build up, and I feel so positive now. (Alex)

It made me feel so wonderful afterwards. I felt like I was walking on air. (Polly)

It was very enjoyable. I found it relaxing and reviving at the same time, if you know what I mean. And thinking and coordinating and everything else which is something you don't practice. (Dora)

…It was a real enjoyable session. I really enjoyed it. …Bearing in mind that I've been stuck in a room in

a hospital for about four or five months. This was an outlet. (David)

I thought it was brilliant. [….] Yes, it really gave me a lift, (Gertie)

Again, participants also talked about the impact on others of taking part and how they had observed people coming to the sessions 'grouchy' and 'grumpy', yet through taking part their demeanour changed significantly and people appeared more cheerful and happier.

and all the other patients were lively and bright too, what they were when they came in. They were all flat and miserable, and they livened up. They were totally different. Yes. They looked totally different when they left. (Mary)

## DISCUSSION

In this study, the dance sessions were perceived to be an activity that provided participants with an opportunity to increase their levels of physical activity. Patients enjoyed the opportunity to interact with others on the ward, and despite some sadness, the majority enjoyed the sessions and experienced a positive effect through taking part.

The perceptions of those interviewed were that the dance sessions were beneficial to their health and well-being. A major strength of this study is that it presents the findings from an innovative programme of music and movement for older people in an acute hospital setting. Semistructured interviews with patients and relatives provide insight into the perceived benefits of the programme, and a rationale for exploring how such programmes could be introduced to increase levels of physical activity and to help alleviate patient boredom on acute hospital wards.

As with all research, there are limitations to this study. The interviews were semistructured and relatively short, not only because they were designed to answer specific questions for the funder but also because participants were frail and easily fatigued. Unstructured interviews may have enabled a more in-depth understanding of the patients' experiences and how these related to their perceptions about the programme. Similarly, data were not collected regarding individuals' functional status or on-going treatments and this may have provided a useful dimension to the research. Furthermore, only patients who participated in groups were interviewed and their responses were all very positive. It would have been interesting to interview those who chose not to attend the sessions and to explore their reasons, particularly as more women than men attended the sessions each week.

The patients participating in the Dance for Health groups were often very elderly (85+) and frail and had been hospitalised for a number of weeks, and so remained seated for the activity. We could find no evidence in the literature on the effectiveness of seated dance as a physical activity for older people, although a systematic review found that dance can have multiple physical benefits for older people, including aerobic power, muscle endurance, strength and flexibility of lower body and balance.[20] Our participants expressed understanding of the need to try and maintain their physical abilities. This resonates with the work of Lafrenière et al who found that older hospitalised adults were cognisant of the risks of functional decline using various strategies to maintain both physical abilities and good spirits.[21] Experiencing things that are familiar and 'normal', such as dance and listening to music, when in an uncertain and stressful environment can also help people to maintain a sense of self. Furthermore, pleasure and enjoyment, a sense of connection, are linked to mental well-being,[22] and social engagement and improving psychological health have been found to lead to positive health outcomes.[23]

It has been recommended that purposeful meaningful activities should be included in multicomponent interventions designed to increase physical activity of patients during a hospital stay.[3] As outlined previously, hospitalised older people can have low levels of activity, which results in deconditioning, and can lead to longer hospital stays, and possible further loss of physical function affecting recovery postdischarge.[24] It is, therefore, imperative that care providers consider alternative forms of exercise such as dance to enable and encourage hospitalised older people to engage in physical activity. A group activity such as dance, which is seen as fun rather than therapy could be an attractive option for patients, particularly as the arts tend to be associated with normative pleasurable experiences making them more attractive, accessible and acceptable than conventional care.[25]

## CONCLUSION

This paper presents the findings from a study of an innovative programme of dance activities introduced onto Elderly Medicine Wards and Stroke and Rehabilitation Unit in an acute hospital trust. As discussed in the introduction, there exists research that demonstrates the impact of dance on the health and well-being of older people in the community and in residential care. The study focused on the experiences of hospitalised older people and their relatives taking part in the dance sessions. It was not possible to conduct a controlled study to look at the impact on the number of falls, whether physical activity increased, or if the rate of rehabilitation improved because of taking part. However, what the study demonstrated is the value placed on the activity by the participants in terms of positive affect and the social opportunity the sessions afforded; in addition, and of significance, is also how the patients recognised the importance of maintaining physical function and how the dance activity encouraged them to move. Further research exploring the use of dance with hospitalised older people would be valuable to investigate whether overall physical activity levels do increase as a result of the dance activity,

with measurement of both physical functioning, and the psychological well-being of participants undertaken.

**Acknowledgements** We would like to thank the patients who were interviewed for this study, and Damian Hebron (Head of Arts), Filipa Pereira-Stubbs (Dance artist), and Deborah Quartermaine (Project Co-ordinator) at Cambridge University Hospitals Trust who were responsible for the delivery of the dance programme.

**Contributors** HB led the study including the design, ethical approval and observed dance sessions and conducted interviews. SH observed sessions and conducted interviews and contributed to the development of the coding framework. Both HB and SH conducted data analysis and data interpretation. HB wrote the first draft of the manuscript and SH reviewed and suggested improvements to the manuscript draft. Both authors contributed to reviewing and editing the manuscript and approved the final manuscript for submission.

**Funding** The Dance for Health Programme was funded by the Dunhill Medical Trust (N187/1116) and included funding for evaluation. The study was commissioned by the Head of Arts via the Addenbrookes Charitable Trust, and the programme delivery team was involved in the study design, but had no input into the data collection, management, analysis or interpretation of the data, review or approvals of the manuscript; or decision to submit the manuscript for publication.

**Competing interests** None declared.

**Patient consent for publication** Not required.

**Ethics approval** As an evaluation involving an intervention in current use meeting the Health Research Authority criteria of an evaluation, the project was registered with the Safety and Quality Support Department at the Cambridge University Hospitals Trust (PRN:6594). It was also approved by the ARU University Faculty Research Ethics Panel (FREP).

**Provenance and peer review** Not commissioned; externally peer reviewed.

**Data availability statement** No data are available. The interview transcripts of the participants were anonymised to maintain confidentiality. Data from this study are not being made available for sharing as participants were not asked for specific consent for this as part of the consent process.

**ORCID iD**
Hilary Bungay http://orcid.org/0000-0001-8202-4521

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
