## [Reviewer comments · BMJ Open]

ARTICLE DETAILS

TITLE (PROVISIONAL)	Older people's perceptions of the impact of Dance for Health sessions in an acute hospital setting: A qualitative study
AUTHORS	Bungay, Hilary; Hughes, Suzanne

VERSION 1 – REVIEW

REVIEWER	Domingo Palacios-Ceña Research Group of Humanities and Qualitative Research in Health Science of Universidad Rey Juan Carlos (Hum&QRinHS), Universidad Rey Juan Carlos, Madrid, Spain.
REVIEW RETURNED	13-Sep-2020

GENERAL COMMENTS	BMJ Open Manuscript Number: bmjopen-2020-044027 Title: Older people and dance in an acute hospital setting: A qualitative study The article is very interesting. But, I think some considerations that will improve the quality of the manuscript. Reviewer used: • Tong A, Sainsbury P, Craig J. Consolidated criteria for reporting qualitative research (COREQ): a 32-item checklist for interviews and focus groups. Int J Qual Health Care. 2007;19(6):349-357. Available at: http://intqhc.oxfordjournals.org/content/19/6/349.long• O'Brien BC, Harris IB, Beckman TJ, Reed DA, Cook DA. Standards for reporting qualitative research: a synthesis of recommendations. Acad Med. 2014;89(9):1245-1251. ABSTRACT: Authors should rewrite results and conclusions section. Many content of the conclusions should be in the results section. Authors must describe total number of participant (not only number of interviews) ARTICLE SUMMARY: STRENGTHS AND LIMITATIONS OF THIS STUDY Authors write: "The first study of the impact of attending a dance group on older people in an acute hospital setting." But Hilary Bungay on of the author participated and published: Bungay H, Hughes S, Jacobs C, Zhang J. Dance for Health: the impact of creative dance sessions on older people in an acute hospital setting. Arts Health. 2020 Feb 6:1-13. doi: 10.1080/17533015.2020.1725072. Epub ahead of print. PMID: 32028845. Authors should modified information at article summary. MAIN TEXT. INTRODUCTION.
---

	Authors must describe “a weekly Dance for Health programme” using the template for intervention description and replication (TIDieR) checklist, and include this material as supplementary material. Hoffmann T., Glasziou P., Boutron I., Milne R., Perera R., Moher D., Altman D., Barbour V., Macdonald H., Johnston M., et al. Better reporting of interventions: Template for intervention description and replication (TIDieR) checklist and guide. BMJ. 2014;348:g1687. doi: 10.1136/bmj.g1687. METHODS. All references that have not been published, and that do not have a doi or ISSN, ISBN number, should be removed. Such as reference 17. Authors should review the coreq standards carefully. Include everything that is not included within the text (http://intqhc.oxfordjournals.org/content/19/6/349.long). Authors must include new sections, and also must include new information: A. Research team and reflexivity section (Interviewer/facilitator, Credentials, Occupation, Gender, Experience and training, Relationship established, Participant knowledge of the interviewer, Interviewer characteristics) B. Participant selection (Sampling, method of approach, sample size and non-participation), C. Setting or context sections. Participants. It is not clear inclusion criteria of older people. All participants with any types of chronic disease were included? And exclusion criteria? How and why research participants were selected; criteria for deciding when no further sampling was necessary. For example, disabling pathologies, functional status during the study, required technical aids, drugs... Context. Authors must describe “a weekly Dance for Health programme” using the template for intervention description and replication (TIDieR) checklist, and include this material as supplementary material. Hoffmann T., Glasziou P., Boutron I., Milne R., Perera R., Moher D., Altman D., Barbour V., Macdonald H., Johnston M., et al. Better reporting of interventions: Template for intervention description and replication (TIDieR) checklist and guide. BMJ. 2014;348:g1687. doi: 10.1136/bmj.g1687. Data collection For a qualitative study using in-depth interviews, it has wide implications that the interviews are so short (5-33 minutes) or so targeted (question guide). The authors should incorporate these aspects into the discussion (study limitations). Regarding interviews: Were questions, prompts, guides provided by the authors? Authors must describe how many participants have participated, how many interviews have been conducted, and (if) how many interviews have been repeated, and why. Authors should describe when and why they stopped interviewing and / or recruiting new participants. Authors must describe who conducted the interviews, where, and who performed the coding process (analysis). Analysis.
--	--

	Authors must describe in detail the coding process used. Include a new table where narratives, codes, categories, and themes are incorporated (coding process elements). Also, authors must include new information: a. Number of data coders (How many data coders coded the data?) b. Description of the coding tree. The sentence: "and was conducted manually to ensure rigour and trustworthiness" (line 23-24) should removed. It is not true that make manual analysis ensure rigour. On the contrary, the process of traceability of the results is ensured with greater precision, using qualitative software. Authors must include new Rigor or trustworthiness section. In this new section authors should include information from analysis section: Page 8, from line 25 to line 44. RESULTS. Authors should include mor information regarding participants. Information related to or that may influence participants' perspectives (dance intervention) should be included. For example, disabling pathologies, functional status during the study, required technical aids. Also, the inclusion criteria should consider these aspects. DISCUSSION. The authors should include at limitations section, aspects such as the short duration of the interviews, the use of a predetermined guide, the small sample, not to incorporate aspects such as functional status, drug use that could influence dancing in the inclusion criteria, etc. References and style journal. Authors should review the BMJ Open journal guidelines. E.i. Use of Vancouver/ Uniform versus APA guidelines. I believe that revision is necessary. I agree to review the manuscript again. Thank you for your work.
--	---

REVIEWER	Kim Dunphy University of Melbourne, Australia
REVIEW RETURNED	24-Sep-2020

GENERAL COMMENTS	I appreciate your engagement wth this important research topic and efforts to offer a quality evaluation. I offer the following specific comments that I would wish to see addressed in a revised version. p. 6, line 42. Mentioning that not all studies have positive outcomes seems a glib comment. Can you explain more about what outcomes came of previous studies in an informative way? p. 6/line 42: I'd suggest referencing these claims: small sample sizes, lack of transparency in the reporting of qualitative data analysis, and heterogeneity of tools and scales used. I note that you repeat two of the weaknesses you identify: heterogeneity of tools and scales used (ie none) and small sample size. Perhaps it might be prudent to comment on weaknesses of previous studies that yours actually addresses. p. 6, line 54: If the philosophy of the program was to be creative and expressive, why did you not consider and assess these as
--

	outcomes? The implied premise here and in the conclusion that a therapeutic intervention would not be creative or expressive is incorrect. Many therapies, including dance movement therapy, place high priority on creativity and expressiveness. p. 8 line 16: 'Data analysis process followed an interpretive approach, and to answer the specific objectives of the study started with pre-defined aims and objectives'. Can you clarify what this means? What were pre-defined aims and objectives? p. 9, line 34: cognitively able seems a slightly clumsy expression p. 10, line 21: 'moving their arms and legs to varying degrees to the music'. Does this mean they didn't move their heads, shoulders, torsos, etc? p. 10, line 8: If the data was gathered systematically and under a structured methodological process it is not anecdotal. You do yourselves a disservice using this term. p. 16, line 29 and again in conclusion: do you mean positive 'effect' or that their 'affect' (the underlying experience of feeling, emotion or mood) was positive? You asked people about their experience of taking part, including questions about their previous experience with dance, timing and other aspects of the program. You don't report these, only outcomes. Why? I see in the data clear description of physical benefit experienced: loosening of muscles and joints, stretching activation. This doesn't correspond at all with your questions re physical outcomes of sleep/appetite and fatigue. Can you offer any insight there? In Article summary, these two points are not strengths or limitations, or you have not explained how they are: An exploration of in-patient experience using semi-structured interviews. The study provides a rationale for investigating how dance in hospital could be used to increase levels of physical activity and to help alleviate patient boredom on acute. Questions: 1. Some of the participants have mentioned that the sessions affect them in a positive way others find them quite demanding what do you think? This seems a poorly worded question, because it implies that being demanding is the opposite of being beneficial. The sessions could actually be beneficial because they are demanding, given the lack of physical demand in every other aspects of these people's lives. This might have had a negative impact on your data collection. Further research is required to investigate the effectiveness of group dance sessions in increasing physical activity on an acute ward, and the potential psychological benefits for hospitalised older people.
--	--

	Wasn't this investigating psychological benefits, given all the emotional and social benefits reported ? I welcome your response.
--	---

VERSION 1 – AUTHOR RESPONSE

Response to Reviewer: 1

Thank you for your thoughtful recommendations as to how we can improve this article. We have done our best to address the points you have made within the confines of the overall word limit for the journal.

ABSTRACT:

Authors should rewrite results and conclusions section. Many content of the conclusions should be in the results section.

The results section and conclusion of the abstract have been revised as recommended.

Authors must describe total number of participant (not only number of interviews)

We have not included this information in the abstract because reviewing the abstracts of recent qualitative papers published in this journal (and others) only the number of participants interviewed in each study are referred to in the abstracts.

ARTICLE SUMMARY: STRENGTHS AND LIMITATIONS OF THIS STUDY Authors write: “The first study of the impact of attending a dance group on older people in an acute hospital setting.” But Hilary Bungay on of the author participated and published: Bungay H, Hughes S, Jacobs C, Zhang J. Dance for Health: the impact of creative dance sessions on older people in an acute hospital setting. Arts Health. 2020 Feb 6:1-13. doi: 10.1080/17533015.2020.1725072. Epub ahead of print. PMID: 32028845.

Authors should modified information at article summary.

The article summary has been revised in line with feedback from both reviewers.

MAIN TEXT.

INTRODUCTION.

Authors must describe “a weekly Dance for Health programme” using the template for intervention description and replication (TIDieR) checklist, and include this material as supplementary material. Hoffmann T., Glasziou P., Boutron I., Milne R., Perera R., Moher D., Altman D., Barbour V., Macdonald H., Johnston M., et al. Better reporting of interventions: Template for intervention description and replication (TIDieR) checklist and guide. BMJ. 2014;348:g1687. doi: 10.1136/bmj.g1687.

A description of the dance activity has been attached as supplementary material. The TIDieR template has not been utilised as it did not seem appropriate to describe the sessions as ‘interventions’ in their current form. We have therefore followed a similar format used by Kier EJP, Lewis A, Williams S, et al. Dance for people with chronic respiratory disease: A qualitative study, BMJ Open 2020; 10e038719.doi.1136/bmjopen-2020-038719, to outline the activity.

METHODS.

All references that have not been published, and that do not have a doi or ISSN, ISBN number, should be removed. Such as reference 17.

Full references now included

Authors should review the coreq standards carefully. Include everything that is not included within the text (<http://intqhc.oxfordjournals.org/content/19/6/349.long>).

Thank you for the suggestions they are very helpful – please see our response to each point below.

Authors must include new sections, and also must include new information:

- A. Research team and reflexivity section (Interviewer/facilitator, Credentials, Occupation, Gender, Experience and training, Relationship established, Participant knowledge of the interviewer, Interviewer characteristics)

We have added details about the research team and the relationship between the participants

- B. Participant selection (Sampling, method of approach, sample size and non-participation),

This section has been updated to justify sample size and the use of purposive sampling.

- C. Setting or context sections.

Information about where the interviews took place and when was already included in the text.

Participants.

It is not clear inclusion criteria of older people. All participants with any types of chronic disease were included? And exclusion criteria? How and why research participants were selected; criteria for deciding when no further sampling was necessary. For example, disabling pathologies, functional status during the study, required technical aids, drugs... Context.

All patients present on the wards who were considered fit enough by the ward staff were invited by the ward staff and dance programme delivery team to participate in the dance sessions. All participants who were able to consent to take part in an interview were eligible to take part. Participants were asked by the nursing staff if they would be willing to speak to the research team no specific inclusion criteria or stratified sampling was applied. We have made revisions reflecting this including inclusion and exclusion criteria to take part in the research.

Authors must describe “a weekly Dance for Health programme” using the template for intervention description and replication (TIDieR) checklist, and include this material as supplementary material. Hoffmann T., Glasziou P., Boutron I., Milne R., Perera R., Moher D., Altman D., Barbour V., Macdonald H., Johnston M., et al. Better reporting of interventions: Template for intervention description and replication (TIDieR) checklist and guide. *BMJ*. 2014;348:g1687. doi: 10.1136/bmj.g1687.

Please see comment above

Data collection

For a qualitative study using in-depth interviews, it has wide implications that the interviews are so short (5-33 minutes) or so targeted (question guide). The authors should incorporate these aspects into the discussion (study limitations).

Thank you for this observation, we agree that in-depth unstructured qualitative interviews would be expected to be longer than those in this study. However, we did not claim to have conducted in-depth interviews, but rather semi-structured interviews were the selected method for this study and the interview length reflects the age of the participants and the context where the interviews took place. The shortest interview was as a result of the patient's visitors arriving which meant the interview was curtailed we have included the data in the study despite the full interview not being concluded.

Regarding interviews: Were questions, prompts, guides provided by the authors?

Yes, we used an interview guide, and this was included as a supplemental file we have now referred to this in the article.

Authors must describe how many participants have participated, how many interviews have been conducted, and (if) how many interviews have been repeated, and why.

The number of interviews conducted is included – we have not included information regarding the data collected as part of the wider study because this is published elsewhere. We also feel to add data from the other strands of research could be confusing without the addition of detail about the other data collection methods used and this would be a problem in terms of the word count for the final article. None of the interviews were repeated.

Authors should describe when and why they stopped interviewing and / or recruiting new participants.

We adopted a pragmatic approach to the sample size, taking account of the range of data collection methods in the study, the potential diversity of participants on the wards where the programme took place, demands on the participants of being interviewed, and the local context. We have revised the text and included a supporting reference.

Authors must describe who conducted the interviews, where, and who performed the coding process (analysis).

This information was included in the original submission.

Analysis.

Authors must describe in detail the coding process used. Include a new table where narratives, codes, categories, and themes are incorporated (coding process elements).

We have included a table demonstrating the coding process (Table 1).

Also, authors must include new information:

- a. Number of data coders (How many data coders coded the data?)

We state that both authors conducted the analysis

- b. Description of the coding tree.

We have included a thematic map (Figure 1)

The sentence: “and was conducted manually to ensure rigour and trustworthiness” (line 23-24) should be removed. It is not true that manual analysis ensures rigour. On the contrary, the process of traceability of the results is ensured with greater precision, using qualitative software.

This was an error – it should have said ‘conducted systematically’ and we have corrected this

Authors must include a new Rigor or trustworthiness section.

In this new section authors should include information from the analysis section: Page 8, from line 25 to line 44.

Although a separate section/headings for trustworthiness and rigour are used in other journals, we can't find any articles in BMJ Open where there is a separate section for rigor and trustworthiness. We have therefore currently left the details within the data analysis section but would be happy to take advice from the editor on this.

RESULTS.

Authors should include more information regarding participants.

Information related to or that may influence participants' perspectives (dance intervention) should be included. For example, disabling pathologies, functional status during the study, required technical aids. Also, the inclusion criteria should consider these aspects.

We did not gather data on functional status or the use of aids. The objective of this strand of the study was to gather participants' perception of the impact of the dance sessions and to support the on-going development of the programme. We have raised this as a limitation in the discussion section.

DISCUSSION.

The authors should include at limitations section, aspects such as the short duration of the interviews, the use of a predetermined guide, the small sample, not to incorporate aspects such as functional status, drug use that could influence dancing in the inclusion criteria, etc.

We have expanded on the limitations of the research in the discussion section. In terms of sample size we acknowledge that there is extensive debate within the literature as to what is an adequate size for qualitative research, but as an exploratory qualitative study in an under-researched area the findings provide an important grounding for future research.

Reviewer: 2

I appreciate your engagement with this important research topic and efforts to offer a quality evaluation. I offer the following specific comments that I would wish to see addressed in a revised version.

p. 6, line 42. Mentioning that not all studies have positive outcomes seems a glib comment. Can you explain more about what outcomes came of previous studies in an informative way?

Thank you for this observation we have added an example from the previous study to illustrate this point.

p. 6/line 42: I'd suggest referencing these claims: small sample sizes, lack of transparency in the reporting of qualitative data analysis, and heterogeneity of tools and scales used. I note that you repeat two of the weaknesses you identify: heterogeneity of tools and scales used (ie none) and small sample size. Perhaps it might be prudent to comment on weaknesses of previous studies that yours actually addresses.

p. 6, line 54: If the philosophy of the program was to be creative and expressive, why did you not consider and assess these as outcomes? The implied premise here and in the conclusion that a therapeutic intervention would not be creative or expressive is incorrect. Many therapies, including dance movement therapy, place high priority on creativity and expressiveness.

The points regarding methodological weaknesses came from a published systematic review and we had omitted the reference, we have addressed this now. Regarding replicating the weaknesses of previous studies –we agree that ideally, we would have set up a study which enabled the use of validated tools and scales to measure outcomes such as balance and cognition. However, the nature of the programme restricted such a study, we did use a structured observation scale which has been used in previous studies of arts and health interventions in hospital settings when observing the sessions. As part of the observations we did collect data on creative engagement with the dance activity to try and capture creative expression (reported in Bungay, Hughes, Jacobs et al 2020). As a sample size for an exploratory qualitative study this is line with the literature, and we feel the findings provide an important grounding which can be used to support further research in this area.

p. 8 line 16: 'Data analysis process followed an interpretive approach, and to answer the specific objectives of the study started with pre-defined aims and objectives'. Can you clarify what this means? What were pre-defined aims and objectives?

The pre-defined aims and objectives were those established by the dance programme delivery team who wanted to understand how patients perceived the impact of the sessions on their subjective wellbeing. However, we can see that the sentence was confusing, and we have revised this sentence for clarity.

p. 9, line 34: cognitively able seems a slightly clumsy expression

We have revised this

p. 10, line 21: 'moving their arms and legs to varying degrees to the music'. Does this mean they didn't move their heads, shoulders, torsos, etc?

Patients did move their whole bodies where they were able to, so we have substituted 'moving their arms and legs' with 'bodies'

p. 10, line 8: If the data was gathered systematically and under a structured methodological process it is not anecdotal. You do yourselves a disservice using this term.

Thank you for this observation we have revised this and removed the word 'anecdotal'

p. 16, line 29 and again in conclusion: do you mean positive 'effect' or that their 'affect' (the underlying experience of feeling, emotion or mood) was positive?

We mean positive affect relating to mood and emotion

You asked people about their experience of taking part, including questions about their previous experience with dance, timing and other aspects of the program. You don't report these, only outcomes.

Why?

Thank you for this - in this article we have only presented the main themes that we identified during the data analysis in relation to the impact on patients taking part in the programme. We could have presented the findings from the questions relating to timing of the programme etc. but decided that the perceived impact on participants would be of more value to the readers rather than the more evaluative aspects of the study.

I see in the data clear description of physical benefit experienced: loosening of muscles and joints, stretching activation. This doesn't correspond at all with your questions re physical outcomes of sleep/appetite and fatigue. Can you offer any insight there?

Question 9 in the interview guide was an open question to participants to try and identify how they perceived the impact of the sessions. The list below was suggested probes to try and elicit information from participants, we did not specifically ask participants about each of these points in turn.

In Article summary, these two points are not strengths or limitations, or you have not explained how they are:

An exploration of in-patient experience using semi-structured interviews.

The study provides a rationale for investigating how dance in hospital could be used to increase levels of physical activity and to help alleviate patient boredom on acute.

We have revised the article summary in line with comments from both reviewers

Questions:

1. Some of the participants have mentioned that the sessions affect them in a positive way others find them quite demanding what do you think?

This seems a poorly worded question, because it implies that being demanding is the opposite of being beneficial. The sessions could actually be beneficial because they are demanding, given the lack of physical demand in every other aspects of these people's lives. This might have had a negative impact on your data collection.

We agree that a physically demanding session could be beneficial to participants but as stated above we were trying to ask an open question to find out what they thought the impact was. We were not trying to suggest to them that they should find it beneficial or demanding, but rather to reassure them that we did not expect or want them to just speak only about the aspects of the sessions that were positive for them.

Further research is required to investigate the effectiveness of group dance sessions in increasing physical activity on an acute ward, and the potential psychological benefits for hospitalised older people.

Wasn't this investigating psychological benefits, given all the emotional and social benefits reported ?

We have reported our findings which indicate participants experienced psychological benefits from the dance activity. These are qualitative findings and in the hierarchy of evidence which is used by funders and commissioners of interventions in health care in the UK greater credibility is given to numerical data provided through validated psychometric tools and scales. The data presented here provides a rationale for further research to undertake pre- and post-test measures to investigate more extensively the psychological impact on older patients on acute wards participating in programmes which contain opportunities for social interaction in a group.